# SOUP-OF-EXPERTS: PRETRAINING SPECIALIST MODELS VIA PARAMETERS AVERAGING

## ABSTRACT

Machine learning models are routinely trained on a mixture of different data domains. Different domain weights yield very different downstream performances. We propose the Soup-of-Experts, a novel architecture that can instantiate a model at test time for any domain weights with minimal computational cost and without re-training the model. Our architecture consists of a bank of expert parameters, which are linearly combined to instantiate one model. We learn the linear combination coefficients as a function of the input domain weights. To train this architecture, we sample random domain weights, instantiate the corresponding model, and backprop through one batch of data sampled with these domain weights. We demonstrate how our approach obtains small specialized models on several language modeling tasks quickly. Soup-of-Experts are particularly appealing when one needs to ship many different specialist models quickly under a model size constraint.

## 1 INTRODUCTION

Large Language Models (LLMs) work well on diverse tasks because they have many parameters and are trained on generalist datasets Brown et al. (2020); Bommasani et al. (2021). However, they are costly to train and to serve, both in terms of memory and inference cost. Specialist language models hold fewer parameters; they are, therefore, cheaper to store, send, and use at inference. However, they must give up the generality of LLMs and specialize in a few specific topics.

In cases where there is an abundance of specialization data, training a small model on those data yields a good specialist. However, in many settings, the specialization data is scarce: for instance, it may come from a narrow topic of interest or be a small company's internal document database. It is, therefore, impossible to train a good-quality specialist model on such data alone.

To get a small model that performs well on the specialization data, we use a large, generic pretraining dataset. That pre-training set contains data from several domains. A powerful method to obtain a good specialist model is importance sampling: it adjusts the mixture weights of the pretraining distribution to resemble the scarce specialist dataset. This method has been shown to outperform generic pre-training (Grangier et al., 2024), but it has a major drawback: it requires pre-training a full model for each specialization dataset available. This makes training cost scale linearly with the number of specialized downstream tasks, which can be intractable as model size and data scales. The goal of this paper is to answer the following question: *How can we leverage a large pre-training set to obtain specialized models that can be instantiated quickly when the specialization data is revealed?* We formalize this question by considering the two phases of serving specialist models.

**Pretraining** We use multiple pre-training domains to train a model. At this point, we do not know the specific data and are unaware of what specific tasks we will need to address later on.

**Specialization phase** We receive a specific dataset, and using the pre-trained model, we need to quickly instantiate a small model that works well on this specific dataset.

In Table 1, we summarize the different costs and constraints associated with these two phases and provide a qualitative review of the strengths and weaknesses of several strategies.

In this landscape of different models, we introduce the Soup-of-Experts, which is designed to be able to instantiate a small specialist model in a flash.

| Model | Spec. size | Pretrain. size | Pretrain. cost | Spec. Cost | Spec. Loss |
|---|---|---|---|---|---|
| Large generic model | Large | Large | Large | Null | Small |
| Mixture of Experts | Large | Large | Small | Null | Med |
| Small generic model | Small | Small | Small | Null | Large |
| Domain Experts | Small | Large | Med | Small | Med |
| CRISP | Small | Null | Null | Large | Small |
| **Soup-of-Experts** | Small | Large | Small | Small | Small |

Table 1: **The different quantities that matter during the phases of serving a specialized model.** Spec. size is the number of parameters in the specialized model. Pretrain. size is the total number of parameters of the pretrained model. Pretrain. cost is the cost of pretraining the model. Spec. cost is the cost to obtain a specialized model when the specialized data is made available. Spec. loss is the loss on the specialized dataset. With these constraints in mind, we compare different models. The goal of this work is to propose the **best possible model under the constraint of having a small specialized model size**. Large generic model is a generalist model, with many parameters, that requires a long training. A mixture of Experts (Fedus et al., 2022; Krajewski et al., 2024) is a small model with added parameters that marginally impact the latency. Since both the LLM and the MoE have many parameters, they are discarded from our study. Small generic model is one small model trained on a generalist distribution. Domain experts (Gross et al., 2017) train one small model per pre-training domain. CRISP (Grangier et al., 2024) trains one model once the specialized data is available using a data mixture that imitates the specialized data distribution. Our proposed method, the Soup-of-Experts, trains one model with many parameters and can quickly instantiate a small model that is good on the specialized data. The results in this table are qualitative.

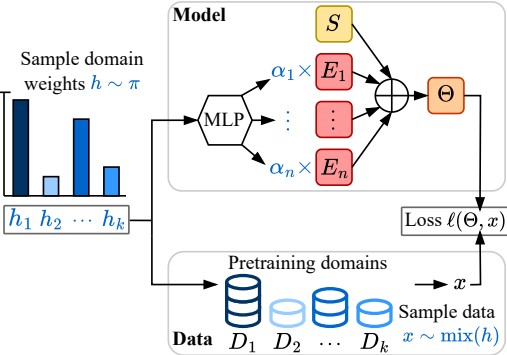

Figure 1: **The Soup-of-Experts and its training pipeline.** The Soup-of-Experts consists of shared parameters $S$, $n$ experts parameters $E_1, \ldots, E_n$, and an MLP that acts as a routing mechanism. At each optimization step, we sample domain weights $h$ from a meta-distribution $\pi$. These domain weights have two purposes: they are passed through an MLP to give a vector of coefficients $\alpha$ that instantiates a **model** by combining the experts' weights, and they are used to sample a mini-batch of **data** following the domain weights law. We then backpropagate through the corresponding loss to update the parameters of the Soup-of-Experts.

Our main idea is to learn to instantiate models with any mixture of domain weights by taking a linear combination of jointly optimized base models, called experts. We are inspired by the works of model merging (Wortsman et al., 2022; Arpit et al., 2022; Rame et al., 2022; 2023; 2024). The gist of model merging is that two model parameters $\Theta_A$ and $\Theta_B$ that are obtained by fine-tuning the same model on different domains $A$ and $B$ can be merged by averaging, yielding a new model $\Theta^* = \frac{1}{2}(\Theta_A + \Theta_B)$, sometimes called a model *soup* (Wortsman et al., 2022), to obtain good performances on both datasets. An important lesson from model merging is that some models' parameters can be linearly combined and yield good models. A caveat of model merging is that the merged models can only be fine-tuned versions of the same base model: for merging to work, the two models must not be too far apart in the parameters space. Our method, Soup-of-Experts, *pre-trains* multiple experts that can, by design, be linearly combined to yield a single specialized model. The linear coefficients of the combination are learned as a function of the pre-training domain weights.

**Paper overview** In Section 2, we explain the details of the Soup-of-Experts, its training pipeline, and how it can be used to instantiate a specialist model quickly. In Section 3, we demonstrate the promises of this approach in a standard language model pre-training setup, where we train small 110M models on Redpajamav2 (Weber et al., 2024) and specialize them on 16 domains from the Pile (Gao et al., 2020).

## 2 METHODS

Figure 1 gives an overview of the proposed architecture, its interplay with data, and its training pipeline. We first explain the training data setup.

**Sampling from the pre-training set** The pre-training set is composed of $k$ domains $D_1, \ldots, D_k \subset \mathcal{X}$ where $\mathcal{X}$ is the sample space (in the case of LLMs, this is the space of text sequences). Each domain contains many samples, usually enough to train a model without repeating data or overfitting. We can query samples from each of these domains, therefore we can sample from a *weighted* mixture of domains: for some **domain weights** $h \in \mathbb{R}^k$, we define the sampling law $\text{mix}(h) = \sum_{i=1}^{k} h_i D_i$ such that $P(x|\text{mix}(h)) = \sum_{i=1}^{k} h_i P(x|D_i)$. This law mixes the datasets $D_i$ with proportions $h_i$, where the domain weights $h$ are non-negative and sum to one. We can efficiently query samples from $\text{mix}(h)$ for any domain weights $h$, by picking a domain $i$ at random following the categorical law induced by $h$, and then sampling an element at random from the corresponding domain $D_i$.

Classical generic pre-training relies on a fixed pre-training domain weights $h_{\text{generic}}$ which define a generic dataset $D_{\text{generic}} = \text{mix}(h_{\text{generic}})$. These weights are defined to train large generalist models that perform well on average. Finding weights for a good average behaviour to train large models is difficult Xie et al. (2023). For smaller models, even a good $\text{mix}(h_{\text{generic}})$ would yield a model far from strong specialists, i.e., giving a model good at everything but excellent at nothing.

**Training with mixtures of pre-training domains** We let $\Theta \in \mathbb{R}^p$ the parameters of a model to be trained on the pre-training set. We define $\ell(\Theta; x)$ the loss function for a sample $x \in \mathcal{X}$ (the next token prediction loss in this paper, since we focus on language modeling). The standard LLM pretraining consists of running Adam (Kingma, 2014) to approximately minimize the **generic loss** $L_{\text{generic}}(\Theta) = \mathbb{E}_{x \sim D_{\text{generic}}}[\ell(\Theta; x)]$. Alternatively, we can train a model on any given mixture with domain weights $h$ by running Adam on the loss $L(\Theta, h) = \mathbb{E}_{x \sim \text{mix}(h)}[\ell(\Theta; x)]$

Grangier et al. (2024) showed that a powerful technique to obtain a good small model on a specific set $D_{\text{spe}}$ is to i) find domain weights $h_{\text{spe}}$ such that $\text{mix}(h_{\text{spe}}) \simeq D_{\text{spe}}$ and then ii) train the model by minimizing $L(\Theta, h_{\text{spe}})$. This importance-sampling-based method called CRISP gives much better specialists than generic pre-training since it trains the model on a distribution that has lots of data and yet is close to the targeted specific distribution.

One caveat of this approach is that it requires retraining a model from scratch anytime one wants to obtain a specialized model. While this cost might be justified in some critical applications, we study alternative avenues to obtain specialized models at a much smaller cost: this is the purpose of the new architecture that we propose in this paper, the Soup-of-Experts.

**Soup-of-Experts** The goal of the Soup-of-Experts is to amortize the training of models on multiple different domain weights. It defines a method that, given training domain weights $h \in \mathbb{R}^k$, quickly instantiates a model $\Theta$ that depends on those domain weights and that yields a low loss $L(\Theta, h)$.

To do so, we enhance the base model with $n$ *experts* $E_1, \ldots, E_n \in \mathbb{R}^p$, which for ease of notation we stack into a matrix $\mathbf{E} = [E_1, \ldots, E_n] \in \mathbb{R}^{n \times p}$. We linearly combine the weights with shared parameters $S \in \mathbb{R}^p$. For a given set of expert coefficients $\alpha \in \mathbb{R}^n$, we instantiate a small model as

$$\Theta = \text{Combine}(S, \mathbf{E}, \alpha) = S + \sum_{j=1}^{n} \alpha_j E_j. \tag{1}$$

Our main idea is to learn coefficients $\alpha$ as a function of the domain weights $h$. To be more precise, we want to learn parameters $S$, $\mathbf{E}$, and a function $\phi : \mathbb{R}^k \to \mathbb{R}^n$ such that, for any domain weights $h \in \mathbb{R}^k$, the instantiated model $\Theta = \text{Combine}(S, \mathbf{E}, \phi(h))$ performs well on the dataset $\text{mix}(h)$,

---

**Algorithm 1** (Grangier et al., 2024) Estimating specialist domain weights that are good for a specialized dataset $D_{\text{spe}}$

---

**Input:** Specialist dataset $D_{\text{spe}}$, embedded domain centroids $c_1, \ldots, c_k$.
**Init:** $h_1, \ldots, h_k = 0$.
**for** $x$ in $D_{\text{spe}}$ **do**
    Find closest centroid: $i = \arg\min \|\texttt{Bert}(x) - c_i\|$
    Increment $h_i = h_i + 1/\#D_{\text{spe}}$
**end for**
**Output:** Domain weights $h_1, \ldots, h_k$

---

i.e., leads to a low loss $L(\Theta, h)$. In practice, we use a two-layer MLP for $\phi$, parameterized by parameters $\omega$, denoted as $\phi_\omega$. Although one can think of many different ways to define a mapping from domain weights to model weights, we chose the parameterization in Equation 1 as it allows us to easily scale the number of total parameters (by increasing the number of experts $n$), and we know from the model merging literature that, perhaps surprisingly, different model parameters can be linearly combined to yield one good model(Wortsman et al., 2022).

**Training Soups of Experts with meta-distributions**  In order to train the Soup-of-Experts to achieve good performance on a diversity of domain weights, we use a **meta-distribution** $\pi$, that is, a sampling law over domain weights. We then train the Soup-of-Experts by minimizing the average error of the model over this meta-distribution, which is the objective function

$$\mathcal{L}(S, \mathbf{E}, \omega) = \mathbb{E}_{h \sim \pi}\left[L(\text{Combine}(S, \mathbf{E}, \phi_\omega(h)), h)\right] \tag{2}$$

We minimize this function using Adam, where at each step, we sample domain weights $h \sim \pi$, instantiate the corresponding model, sample a mini-batch from $\text{mix}(h)$, and do an optimization step on $\ell(\text{Combine}(S, \mathbf{E}, \phi(h)), x)$. Figure 1 illustrates this training pipeline.

The choice of meta-distribution $\pi$ has a critical role on the Soup-of-Experts. Ideally, it should reflect the distribution of specific tasks that one wishes to address during the specialization phase. In our experiments, we favor sparse domain weights and use meta-distributions $\pi$ that first sample $s \ll k$ domains and then take uniform random domain weights over these $s$ domains.

**Instantiating a Soup-of-Experts: Specialization in a flash**  After pre-training, the Soup-of-Experts has the flexibility to quickly provide a model that is good for any data distribution domain weights $h$, simply by forming the parameters $\Theta = \text{Combine}(S, \mathbf{E}, \phi(h))$. This instantiation only requires a forward pass through a small MLP, and merging $n$ parameters; it does not require any training. To specialize a Soup-of-Experts, we obtain domain weights $h_{\text{spe}}$ from $D_{\text{spe}}$ so that $D_{\text{spe}} \simeq \text{mix}(h_{\text{spe}})$. To do so, we use the nearest-neighbor method of (Grangier et al., 2024), which is described in Algorithm 1 for completeness. We then instantiate the parameters $\text{Combine}(S, \mathbf{E}, \phi(h_{\text{spe}}))$. This model can then be fine-tuned on the specialization data to increase its performance if the computational budget allows it.

## 3 EXPERIMENTS

We first detail the experimental setup: datasets, models, metrics, and hyperparameters.

**Pretraining domains** We pre-train language model on Redpajama2 (Weber et al., 2024), a widely used curated web-crawl dataset. We obtain the pre-training domains $D_1, \ldots, D_k$ with the same clustering method as Grangier et al. (2024): we embed each document using sentence-bert (Devlin, 2018), and then use the k-means algorithm on these embeddings to split the dataset into $k$ pre-training domains. We use a hierarchical k-means, where we first cluster the dataset into $k = 64$ domains and then cluster each of these domains into $64$ smaller domains, yielding in total $k = 4096$ domains. We also collect the $k$ corresponding centroids $c_1, \ldots, c_k$ in the embedding space, in order to use Algorithm 1 to obtain specialist domain weights.

**Specialization domains** We consider 16 datasets from the PILE (Gao et al., 2020) as target specialization sets: arxiv, dm_mathematics, enron emails, europarl, freelaw, github, hackernews, nih exporter, openwebtext, pg19, phil papers, pubmed, stackexchange, ubuntu, uspto, and wikipedia.

For each of these datasets, we compute the corresponding specialist domain weights using Algorithm 1. We evaluate different methods on each of the specialization datasets individually, and we report averaged losses over these domains. We defer individual domain results to the appendix. We highlight that these specialist domains and specialist domain weights are never used or seen during the pre-training phase for all methods except for CRISP.

**Models** We consider GPT-2 type transformer architectures, which we train with the next-token-prediction loss. We consider a base model size of $110M$ parameters.

**Metrics** In this work, we measure the ability of a model on a specialization dataset with its next-token prediction loss on that domain: we focus solely on language modeling. This loss predicts well the downstream performance of models with more complex metrics like reasoning, question-answering or translation ability (Gonen et al., 2022; Du et al., 2024; Gadre et al., 2024).

**Training hyperparameters for the Soup-of-Experts** Unless specified otherwise, we train the Soup-of-Experts with $n = 128$ experts. With a base model size of $110M$, these Soup-of-Experts therefore hold a total of $(128 + 1) \times 110M = 14B$ parameters, that can be linearly combined into small $110M$ models. We use a meta-distribution $\pi$ with a support size of $s = 4$.

All the methods we compare in this work instantiate, at specialization time, a model with the same architecture and number of parameters. As explained in Table 1 we consider the following models:

**Generic Pretraining** We train one generic model on the standard pre-training distribution. At specialization time, the model stays the same and is evaluated on the specialization set.

**Domain experts** (Gross et al., 2017) We train one model on each pretraining domain $D_i$. At specialization time, we select the model that yields the smallest loss on the specialization set. This technique does not scale with the number of domains. We only train $k = 64$ domain experts, as it would be infeasible to train 4096 with our budget.

**CRISP** (Grangier et al., 2024) We train one model per specialization set on the mixture $\mathrm{mix}(h_{\mathrm{spe}})$. At specialization time, we use the corresponding model. This method does not scale with the number of specialization domains; it requires one pre-training run per specialization domain.

**Main results** We report the training curves on the pre-training set as well as the average loss on the specialization domains in Figure 2. The specialized loss is obtained by computing the loss on each specialization domain for the corresponding specialist domain; each domain uses a different model (except for the generic pretraining method, which uses the same model for each specialization set).

The x-axis corresponds to time, which in this case is close to being proportional to the computational cost required to train the model (indeed, the cost of instantiating the experts is small in front of that of backpropagating through the network; we get a throughput with the SoE that is $77\%$ of that of the generic pretraining).

For the Soup-of-Experts and the generic pre-trained models, the training time is unambiguous. For the two other baselines, which train multiple models, we report the total training time taken by all the models.

We observe that the Soup-of-Experts achieves the best performance among all methods on the specialized domains, and is only slightly worse than generic pre-training on the pre-training loss (while generic pre-training explicitly minimizes this loss).

The Soup-of-Experts and the generic pretraining are the only scalable methods with respect to the number of pretraining domains and number of specialization domains. Indeed, we consider 16 specialization domains here. Had we considered more domains, the CRISP method would have taken more and more pre-training time. Similarly, increasing the number of domains would increase the computational cost of the domain experts method a lot.

**Complementarity to fine tuning** For each of the pile domains, we instantiate the corresponding Soup-of-Experts. We then fine-tune this model and the baseline model with different numbers of available fine-tuning tokens. We report the validation losses in Figure 2, as well as the number of tokens the generic pretraining method needs to use to recover a performance similar to that of the Soup-of-Experts.

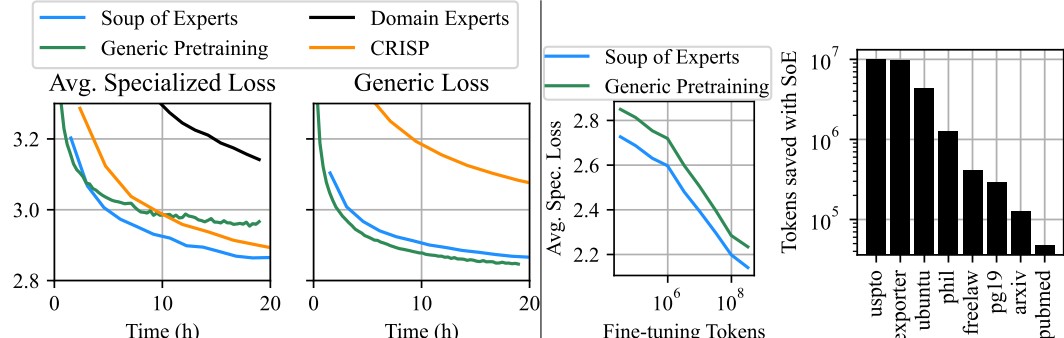

Figure 2: **Left: training curves of the different methods.** The average specialized loss is the average of the loss of the models over 16 domains from the Pile. The generic loss is the loss of the models on the standard pre-training distribution of RedPajamav2. The x-axis is the training time. This number is roughly proportionnal to number of tokens processed, since in this setting, the cost of instantiating the Soup-of-Experts is small in front of that of backpropagating through the network. The domain experts and CRISP have to train many models, so they are not competitive in this setup. The Soup-of-Experts performs almost similarly to generic pre-training on the generic loss, which means that it holds the general knowledge in the pre-training set, while CRISP and Domain Experts are not good generalists (Domain Experts are even out of the figure limits on the right figure). The Soup-of-Experts gives the best specialists, as seen on the left figure. **Right: The gains of Soup-of-Experts during pretraining are maintained during fine-tuning and sometimes lead to large savings.** On each of the 16 domains from the PILE, we fine-tune the corresponding instantiated Soup-of-Experts and generic model, with a limited number of fine-tuning tokens. We stop fine-tuning at the point where validation loss starts increasing. **Left:** Average loss over domains. We see that the Soup-of-Experts maintains its advantage regardless of the number of available fine-tuning tokens. **Right:** The number of fine-tuning tokens one needs to fine-tune the generic model to reach the same validation loss as the *base*, not fine-tuned, Soup-of-Experts. For example, on `uspto`, one needs $10M$ tokens to fine-tune the generic model and reach the same loss as the Soup-of-Experts instantiated on `uspto` out of the box after pre-training.

## CONCLUSION

We have introduced a novel asymmetrical architecture, the Soup-of-Experts. It holds a large set of expert parameters that encodes a family of small, stand-alone models obtained by linear combination of the parameters. We propose a learning algorithm so that the coefficients of the linear projection are a function of the domain weights from which the input is sampled. A pre-trained Soup-of-Experts can, therefore, instantiate instantly a model tailored to any mixture of domain weights. We demonstrated the benefits of this approach on standard datasets, even when these datasets and the corresponding domain weights are unavailable when the soup is trained.

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
