# OpenReview forum: "Soup-of-Experts: Pretraining Specialist Models via Parameters Averaging"
_ICLR.cc/2025/Workshop/MCDC — MCDC @ ICLR 2025_

### Official Review · Reviewer_mK5j · 2025-02-28

**Rating:** 8
**Confidence:** 4
**Fit:** 5

**Summary:**

The paper proposes Soup-of-Experts (SoE), the method which pretrains a set of expert parameters that can be linearly combined to instantiate small, specialized language models without extensive retraining. Using a combination of bank of experts, shared parameters, and a MLP router they obtain rapid specialization, while being competitive to the generic pretraining setup. Evaluated on popular and established datasets, their method outperforms other baselines in the specialised setting while maintaining good general performance.

**Reason For Giving A Higher Score:**

SoE tackles a timely challenge with an elegant, modular design and promising results. Addressing the meta-distribution’s role and specialization sample size would strengthen its claims, but these gaps don’t overshadow its innovation or potential.

**Reason For Giving A Lower Score:**

Lack of ablations and some experimental details.

**Strengths And Weaknesses:**

Strengths:
- SoE leverages parameter averaging to create a modular, scalable solution for model specialization, aligning with the workshop’s focus on reusable ML components.
- The method excels in specialized domains and reduces fine-tuning costs, as shown in Figure 2.

Weaknesses:
- Authors choose sparse meta distributions, and don't explore the motivation deeply or test alternative meta distribution choices. The following question remains; How do these affect training and performance. Moreover, robustness to this choice remains unaddressed.
- Algorithm 1 estimates the combination of experts from specialization data, but the number of samples used is unspecified. This omission obscures how robust SoE is to limited data compared to baselines—key for practical deployment where specialization samples may be scarce.

**Suggestions:**

- Analyse the impact of different choices of meta-distribution during training.
- Report sample size and test performance with varying amounts of specialization data.

---

### Official Review · Reviewer_NSZr · 2025-03-02

**Rating:** 6
**Confidence:** 4
**Fit:** 4

**Summary:**

This paper introduces a method for sampling a specialized model from a warehouse of expert models with the aid of a meta-domain distribution vector. The method shines in scenarios requiring rapid deployment of small, specialized models under size constraints, such as language modeling tasks. By avoiding retraining for each domain, Soup-of-Experts achieves efficiency and flexibility, delivering low loss $L(\theta, h)$ for the instantiated models tailored to specific domain needs.

**Reason For Giving A Higher Score:**

N/A

**Reason For Giving A Lower Score:**

N/A

**Strengths And Weaknesses:**

**Strengths**:
- The motivation of the paper is appealing and practical.
- The training pipeline is novel and interesting.
- The qualitative results are promising.

**Weakness**:
- I believe the author should also consider comparing the proposed method with other fast model merging baselines, such as [1] and [2]. If these baselines are not suitable for comparison, I recommend combining the proposed method with them to see if they can boost each other's performance.
- The experiments are conducted only on GPT2-base model size. I am wondering if the proposed method worked well for both small and large model sizes.

[1] Prateek Yadav, Derek Tam, Leshem Choshen, Colin Raffel, and Mohit Bansal. 2023. **TIES-Merging: Resolving Interference When Merging Models.** In *Neurips 2023*.

[2] Le Yu, Bowen Yu, Haiyang Yu, Fei Huang, and Yongbin Li. 2024. **Language Models are Super Mario: Absorbing Abilities from Homologous Models as a Free Lunch**. In *ICML 2024*.

**Suggestions:**

1. Add more baselines comparison.
2. Add more information about the model's parameters number and an ablation on different model scale.

---

### Official Review · Reviewer_KrcX · 2025-03-02

**Rating:** 7
**Confidence:** 4
**Fit:** 4

**Summary:**

The paper proposed a new scheme for creating a small specialized model from a larger one without requiring any fine-tuning. This specialized model is targeted toward a specific domain following a specific distribution of the pretrained domains, modelled by domain weights $h$. The specialized model can further be seen as the good initialization, and then fine-tuned on the target domain to get better performance with fewer steps.

**Reason For Giving A Higher Score:**

N/A

**Reason For Giving A Lower Score:**

N/A

**Strengths And Weaknesses:**

**Strength:**
- The idea is straight-forward and easy to follow.
- The authors provide an efficient training pipeline for learning the mapping function between the domain weights $h$ and the experts parameters $\alpha$.
- The specialized model can be created beforehand, thus lowering the computational and memory cost for deploying model in practice.

**Weakness:**
- The quantitative result is missing, only loss is provided. I would like to see more meaningful metrics like perplexity or other downstream tasks' metric.
- No direct comparison with others model merging methods, as in zero-shot setting as well as in fine-tuning setting.
- While this idea is interesting, is there anyway to used pretrained models instead of training the model from scratch?

**Suggestions:**

See above section.

---

### Official Review · Reviewer_dKCr · 2025-03-05

**Rating:** 7
**Confidence:** 4
**Fit:** 5

**Summary:**

This paper proposes a pretraining framework that leverages, "trainable" parameter averaging across experts while all of the experts are being trained. After the pretraining, an estimate of domain weight distribution for the target data (they use the nearest-neighbor method) is sufficient to compute interpolation weights and use for parameter averaging. More concretely, a simple two-layer MLP is used to project domain weights to interpolation weights (which is trained end-to-end during pretraining).

The paper compares the proposed method with naive pretraining and CRISP where importance sampling is used to pretrain strong specialized models. While CRIPS requires to pretrain a new model for each domain of interest, the proposed techniques only requires lightweight meta-training to learn interpolation weights. The paper shows promising results, showing the best performance on specialized domains while being slightly worse than naive pretraining on the pretraining loss.

**Reason For Giving A Higher Score:**

I refer to my comments above about the "strengths" of the work as reasons for a high score.

**Reason For Giving A Lower Score:**

I refer to my comments above about the "weaknesses" of the work as reasons for a low score.

**Strengths And Weaknesses:**

Strengths :
1. Soup-of-expert is a very novel framework that leverages parameter averaging in the pretraining time, leading to better performance in the inference time on specialized domains with a smaller model size. Especially in the inference time, the framework does not require any training for computing interpolation weights except domain embeddings for their centroids.

2. The paper is very well written, and the framework is described clearly. In this context, domain sampling, learning interpolation weights, and pretraining experts include sufficient details.

3. Experimental results are promising.

Weaknesses:
1. I think the main weakness of the paper is the baselines and comparison in their experiments. Although the proposed method averages the parameters in the inference time, leading a small model (110M parameters in the experiments), it pre-trains many more parameters (14B). Instead of comparing independently trained models (standard pretraining or CRISP), comparing SoE to an MoE architecture with Top-1 expert (with adjusted parameters to make total and active parameter count similar) would be more fair.

2. Similarly, in the experiments, SoE was trained with 1+128 experts, however, Domain expert was trained with 64 experts (64 separate models), which leads to an unfair comparison.

**Suggestions:**

As mentioned above, I suggest the authors include a comparison with MoE-based architecture. More concretely, these 2 architectures would be best to compare:
1. Standard sparse MoE with Top-1 expert where the dimensions are adjusted to make the total and active number of parameters the same with the proposed method.

2. Branch-Train-Merge (https://arxiv.org/abs/2208.03306), where a base model is first pre-trained, then new branches are trained/finetuned to merge or ensemble in the inference time.

---

### Official Review · Reviewer_rwsU · 2025-03-05

**Rating:** 6
**Confidence:** 5
**Fit:** 3

**Summary:**

This paper proposes a new method, namely,  Soup-of-Experts, that pre-trains multiple experts that are inherently designed to be linearly combined, resulting in a single specialized model.

**Reason For Giving A Higher Score:**

I give a marginally above acceptance threshold score for this paper since the use of model merging in a meta-learning setting is an interesting direction to explore.

**Reason For Giving A Lower Score:**

N/A

**Strengths And Weaknesses:**

Strength: The paper explores a new use of model merging in a meta-learning setting.

Weakness:

1. The paper is not quite novel, which is a combination of existing techniques.

2. Experimental results are not sufficient, and much more will be needed to justify the advantages of the proposed method.

3. The presentation of the paper needs to be enhanced.

**Suggestions:**

1. Experiments on different tasks and with larger models will help improve the paper.

2. Providing theoretical guarantees for the proposed method will be a plus.

---

### Decision · Program_Chairs · 2025-03-06

**Decision:**

Accept

**Comment:**

This work proposes a soup of experts where a model can be created from a specific merging. This is a relevant topic to this workshop, and all reviewers recommend acceptance, therefore we're please to accept this work to the workshop.